# Atypical Clinical Presentation of Laryngopharyngeal Reflux: A 5-Year Case Series

**DOI:** 10.3390/jcm10112439

**Published:** 2021-05-31

**Authors:** Jerome R. Lechien, Stéphane Hans, Francois Bobin, Christian Calvo-Henriquez, Sven Saussez, Petros D. Karkos

**Affiliations:** 1Department of Human Anatomy and Experimental Oncology, Mons School of Medicine, UMONS Research Institute for Health Sciences and Technology, University of Mons (UMons), B7000 Mons, Belgium; sven.saussez@umons.ac.be; 2Department of Otolaryngology-Head & Neck Surgery, Foch Hospital, School of Medicine, UFR Simone Veil, Université Versailles Saint-Quentin-en-Yvelines (Paris Saclay University), 92150 Paris, France; prhans.foch@gmail.com; 3Department of Otolaryngology-Head & Neck Surgery, Ambroise Paré Hospital, APHP, Paris Saclay University, 92150 Paris, France; 4Department of Otolaryngology-Head & Neck Surgery, CHU Saint-Pierre, Faculty of Medicine, University Libre de Bruxelles, B1000 Brussels, Belgium; 5Elsan Polyclinic of Poitiers, 86000 Poitiers, France; bobinfr@wanadoo.fr; 6Department of Otolaryngology, Hospital Complex of Santiago de Compostela, 15700 Santiago de Compostela, Spain; christian.calvo.henriquez@gmail.com; 7Department of Otolaryngology-Head & Neck Surgery, EpiCURA Hospital, B7000 Mons, Belgium; 8Department of Otorhinolaryngology and Head and Neck Surgery, AHEPA University Hospital, Thessaloniki Medical School, 54621 Thessaloniki, Greece; pkarkos@aol.com

**Keywords:** reflux, laryngopharyngeal, clinical, atypical, nasal, otological, respiratory, management, treatment, diagnosis

## Abstract

Background: Laryngopharyngeal reflux (LPR) is a common disease in otolaryngology characterized by an inflammatory reaction of the mucosa of the upper aerodigestive tract caused by digestive refluxate enzymes. LPR has been identified as the etiological or favoring factor of laryngeal, oral, sinonasal, or otological diseases. In this case series, we reported the atypical clinical presentation of LPR in patients presenting in our clinic with reflux. Methods: A retrospective medical chart review of 351 patients with LPR treated in the European Reflux Clinic in Brussels, Poitiers and Paris was performed. In order to be included, patients had to report an atypical clinical presentation of LPR, consisting of symptoms or findings that are not described in the reflux symptom score and reflux sign assessment. The LPR diagnosis was confirmed with a 24 h hypopharyngeal-esophageal impedance pH study, and patients were treated with a combination of diet, proton pump inhibitors, and alginates. The atypical symptoms or findings had to be resolved from pre- to posttreatment. Results: From 2017 to 2021, 21 patients with atypical LPR were treated in our center. The clinical presentation consisted of recurrent aphthosis or burning mouth (*N* = 9), recurrent burps and abdominal disorders (*N* = 2), posterior nasal obstruction (*N* = 2), recurrent acute suppurative otitis media (*N* = 2), severe vocal fold dysplasia (*N* = 2), and recurrent acute rhinopharyngitis (*N* = 1), tearing (*N* = 1), aspirations (*N* = 1), or tracheobronchitis (*N* = 1). Abnormal upper aerodigestive tract reflux events were identified in all of these patients. Atypical clinical findings resolved and did not recur after an adequate antireflux treatment. Conclusion: LPR may present with various clinical presentations, including mouth, eye, tracheobronchial, nasal, or laryngeal findings, which may all regress with adequate treatment. Future studies are needed to better specify the relationship between LPR and these atypical findings through analyses identifying gastroduodenal enzymes in the inflamed tissue.

## 1. Introduction

Laryngopharyngeal reflux (LPR) may be defined as an inflammatory condition of the upper aerodigestive tract with tissues related to the direct and indirect effect of gastric or duodenal content reflux, inducing morphological changes in the upper aerodigestive tract [1]. Gastroesophageal reflux disease (GERD) and LPR share some common pathophysiological mechanisms but may differ regarding the nature and time of occurrence of reflux events [1]. Many basic science studies have demonstrated that the mucosal lesions are mainly due to the extra- or intracellular pepsin activity into the upper aerodigestive tract mucosa [2,3]. Pepsin was found in the nasal mucosa of patients with resistant chronic rhinosinusitis and LPR [4]. Others identified LPR as a key condition responsible of nasal symptoms in patients who do not report sinonasal infection [5]. In the same way, pepsin and LPR were identified as important factors in the development of chronic otitis media in children and adults [6], laryngeal disorders [7], or bronchial irritation in patients with asthma [8]. The involvement of LPR in many respiratory and digestive conditions may lead to atypical clinical presentation of the disease, which may be difficult to detect in clinical practice.

This paper attempts to present a case series of patients with atypical clinical presentations of LPR diagnosed in our reflux clinic.

## 2. Methods

### 2.1. Design, Data Collection, and Setting

A retrospective medical chart review of patients who were diagnosed with LPR from 2017 to 2021 at the European Reflux Clinics (Brussels, Paris, Poitiers) [9] was performed. The LPR diagnosis was made through 24 h hypopharyngeal-esophageal multichannel intraluminal impedance-pH monitoring (HEMII-pH) respecting predefined criteria in patients who initially reported LPR-related symptoms, e.g., hoarseness, dysphagia, throat pain, throat clearing, halitosis, or globus sensation [10].

The atypical LPR was defined as a clinical presentation with symptoms or findings that are not reported in the reflux symptom score (RSS) [11] or reflux sign assessment (RSA) [11]. RSS is a 22-item patient-reported outcome questionnaire that reports the most prevalent otolaryngological, digestive, and respiratory symptoms associated with LPR. RSA is a finding instrument including the most prevalent signs associated with LPR. Thus, the development of the RSA was based on an initial observational study analyzing the prevalence of oral, laryngeal, and pharyngeal findings associated with LPR in patients with a confirmed diagnosis (HEMII-pH). Both scores were developed after a systematic review of the most prevalent LPR symptoms and signs reported in the literature [12] and may be considered as a complete, reliable, and validated patient-reported outcome questionnaire or finding instrument [11].

The association between atypical findings and LPR was confirmed if the LPR diagnosis was confirmed with HEMII-pH, if the finding resolved posttreatment, and if the atypical finding of the patient was not explained by another condition. Rigorous exclusion criteria were subsequently used to select well-matched samples, to minimize bias, and to eliminate confounding factors. Patients with other comorbidities different from LPR or gastroesophageal reflux disease (GERD) such as smoking, drinking, or an active allergy at the time of the evaluations were excluded. Incomplete medical records were also excluded.

The epidemiological, medical, and therapeutic data of each patient who consulted in our center were all recorded, electronically available in our system, and were easily extracted for the purpose of the study using the following keywords: “atypical”, “unusual”, “uncommon”, “nasal”, “respiratory”, “bronchial”, “ear”, and “eye”.

### 2.2. 24 h HEMII-pH

The HEMII-pH catheter was composed of 8 impedance ring pairs and 2 pH electrodes (Versaflex Z^®^, LPR ZNID22+8R FGS 9000-17; Digitrapper pH-Z testing System, Medtronic, Hauts-de-France, Lille, France, Supplementary file). The catheter model used was introduced transnasally and chosen based on the esophageal length of the patient. Six impedance segments were placed along the esophagus zones (Z1 to Z6) below the upper esophagus sphincter (UES). Two additional impedance segments were placed 1 and 2 cm above the UES in the hypopharyngeal cavity. The configuration of this catheter enabled the recording of changes in intraluminal impedance at each point. The two pH electrodes were placed 5 cm above the LES and 1–2 cm above the UES. The HEMII-pH probe was placed in the morning before breakfast (8:00 A.M). A hypopharyngeal reflux event (HRE) was defined as an episode that reached two hypopharyngeal impedance sensors. A LPR diagnosis was given if there was ≥1 acid or nonacid HRE [13]. Acid reflux was defined as an episode with pH ≤ 4.0. Nonacid reflux consisted of an episode with pH > 4.0. The HEMII-pH tracing was electronically analyzed by the software and the result was verified by two senior physicians. LPR was defined as acid when the ratio of the number of hypopharyngeal acid reflux episodes/number of nonacid reflux episodes was >2. LPR was defined as nonacid when the ratio of the number of acid reflux episodes/number of nonacid reflux episodes <0.5. Mixed reflux consisted of a ratio ranging from 0.51 to 2.0. GERD was defined as a DeMeester score >14.72 or a length of time >4.0% of the 24 h recording spent below pH 4.0.

### 2.3. Finding Evolution, Treatment, and Management of Atypical Clinical Presentations

The management of patients in our reflux clinics is summarized in Figure 1. In practice, after the HEMII-pH diagnosis, the laryngologists started a certain treatment depending on the HEMII-pH features. The treatment scheme included a diet, behavioral changes, and the use of proton pump inhibitors (PPIs), alginate, or magaldrate for 3 months. PPIs were taken once or twice daily before meals depending on the pattern of reflux events (daytime, nighttime reflux events). Alginates were taken twice or three times daily after the main meals in the case of weakly acid (mixed) or nonacid LPR. Diet recommendations were based on a validated European diet scheme [14]. The treatment of patients was custom-tailored at 3 and 6 months regarding the evolution of RSS.

Nonresponders or those presenting with an unusual clinical presentation benefited from additional general and specific (related to the anatomical findings) examinations in order to identify a differential diagnosis or comorbidities associated with LPR (Figure 2).

## 3. Results

From the 351 patients who had a positive HEMII-pH diagnosis, 24 patients met our inclusion criteria. Three patients were excluded because there were no posttreatment data in the medical record. The atypical findings consisted of recurrent aphthosis or burning mouth (*N* = 9), recurrent burps and abdominal disorders (*N* = 2), posterior nasal obstruction (*N* = 2), recurrent acute suppurative media otitis (*N* = 2), severe vocal fold dysplasia (*N* = 2), recurrent acute rhinopharyngitis (*N* = 1), chronic tearing (*N* = 1), recurrent aspirations (*N* = 1), and tracheobronchitis (*N* = 1). The patient features are reported in Table 1, Table 2 and Table 3.

### 3.1. Oral Atypical Manifestations

Eleven patients reported oral atypical manifestations. From them, two patients had recurrent aphthosis and burning mouth syndrome, while seven individuals had severe isolated burning tongue/mouth. Before submitting them to HEMII-pH, the patients benefited from complete dental and maxillofacial examinations, excluding the following lesions or conditions associated with secondary burning mouth syndrome: atrophic glossitis, geographical tongue, other aphthosis causes, dysplasia, lichen, mycosis, Sjogren, autoimmune disease, vitamin disorders, or hypersensitivity to dental materials. After the exclusion of these causative factors, they benefited from a reflux consultation and a 24 h HEMII-pH. As exhibited in Table 1, LPR was identified in all patients, consisting of acid (*N* = 7), weakly acid (*N* = 1), and alkaline (*N* = 1) LPR. Regarding the HEMII-pH features, patients received a personalized treatment and the disorders/lesions regressed after a 3- to 6-month therapy. There was no recurrence of the disorder at the last f time, ranging from 6 months to 3 years. Note that patient number 6 also developed fissured tongue (Figure 3A), which did not change after treatment.

Patient number 3 was living abroad and was referred to our clinic with severe anorexia related to burning mouth syndrome resistant to 3- to 6-month anti-reflux therapy (i.e., the use of PPIs, alginate, and an antireflux diet). The patient lost 15 kg over the previous 6 months. The HEMII-pH revealed alkaline LPR, and the patient was treated with magaldrate (four times daily) for 6 months. As the symptoms did not improve, an additional check-up was proposed to the patient and a histamine intolerance was detected. The symptoms and findings disappeared after 2 months of a histamine-free diet.

Among the patients with oral findings, two patients complained of recurrent burps, halitosis, and abdominal pain. As they were resistant to PPI therapy, patients were referred to our specialized clinic. LPR diagnosis was confirmed with the HEMII-pH, and the digestive work-up (biology and lactose hydrogen breath test) revealed gluten (patient n10) and lactose (patient n11) intolerance. The gluten-free and lactose-free diets were sufficient to significantly improve laryngopharyngeal and digestive symptoms in these patients over the long-term follow-up (4 years).

### 3.2. Otological and Nasal Atypical Manifestations

Six patients had otological or nasal atypical LPR presentations (Table 2). Among them, three individuals reported resistant chronic nasal obstruction, which was not related to a nasal or nasopharyngeal tumor, polyposis, chronic rhinosinusitis, septal deviation, allergic rhinitis, inflammatory nasal disease, cartilage hypotonia, infection, or chemical- or drug-induced rhinitis. A CT scan of the nose and sinuses was unremarkable. There was no history of nasal surgery and they did not respond to a 3-month topical treatment including saline solution irrigation and two different corticosteroids (mometasone furoate and budesonide). The patients benefited from acoustic rhinomanometry to confirm the nasal obstruction, which was related to inferior turbinate hypertrophy. In patient number 13, a turbinate edema was located in the back of the turbinate. The RSS and the nasal obstruction of patients significantly improved after a 3- to 6-month antireflux therapy. Two patients were weaned from the antireflux medication and were clinically controlled with the antireflux diet over the long-term. One patient was not weaned from the alginate-based treatment, because she continued to have laryngopharyngeal symptoms (LPR chronic course). At baseline, this patient also had chronic tearing related to an inferior meatus edema. Although the laryngopharyngeal symptoms persisted, the inferior meatus edema and the related tearing disorder disappeared.

Three patients had an otological clinical presentation associated with LPR (Table 2). Patient number 15 had recurrent acute suppurative otitis media (3 to 4 times annually) throughout the last decade. During the last episode, the otolaryngologists observed bulging and erythema of both tympanic membranes and the patient benefited from antibiotic/anti-inflammatory treatment. The predisposing factors for recurrent otitis media were all excluded (e.g., immunological disorders, nasal disorder, rhinitis, and chemical exposure). Patient number 16 also reported otological disorder (retraction pocket) without history or favoring factors. These two patients were addressed to the reflux clinics by general otolaryngologists who observed LPR-related signs and erythema of the nasopharyngeal cavity (Figure 3B). Acid gaseous upright and daytime LPR was confirmed in both cases. The personalized treatments led to a complete resolution of the recurrent acute otitis media history and retraction pocket after treatment. There was no recurrence at one year posttreatment. The last patient had a chronic course of rhinopharyngitis with severe rhinorrhea, postnasal drip, nasal obstruction, and face and ear pressure. Nasal fiberoptic endoscopy revealed significant nasopharyngeal sticky mucus and erythema (Figure 3C). The CT scan and otological examinations (i.e., otoscopy, tympanometry, and audiometry) were unremarkable. There was also a dust allergy that was controlled by antihistamines. As for the other patients, the HEMII-pH confirmed the diagnosis, and the rhinopharyngitis-related symptoms disappeared after a personalized treatment.

The third patient group included two individuals with severe vocal fold dysplasia, one with recurrent aspirations and related lung infections, and another with recurrent tracheobronchitis. No patient smoked or had tobacco or chemical exposure history. Prior to the reflux consultation, patients with vocal fold leukoplakia underwent a direct laryngobronchoscopy with a biopsy confirming severe dysplasia (Figure 3D). Patients with aspiration and tracheobronchitis benefited from a complete pulmonary work-up, including lung spirometry, bronchoscopy, and chest CT-scan, which were normal. The patient with aspiration had no neurological disorder, and videofluoroscopy and bronchoscopy were unremarkable. HEMII-pH identified acid or weakly acid LPR in these patients. The patient disorders disappeared with the personalized treatment and there was no recurrence over the follow-up period.

## 4. Discussion

Laryngopharyngeal reflux is occasionally associated with nonspecific symptoms and findings, which make diagnosis challenging for unaware physicians [15]. The involvement of LPR in the development of several inflammatory conditions of the upper aerodigestive tract has increasingly been studied over the past decades, reporting potential involvement in rhinological, otological, and laryngological diseases [5,6,7,8]. In this study, our team shared some clinical observations where the diagnosis and the treatment of LPR disease had a significant impact on the resolution of specific conditions that are currently not or poorly known to be associated with reflux.

The involvement of LPR in the development of oral disorders was suspected for a long time, the first reports dating from the 1970s [16]. In the present study, we reported several patients with primary burning mouth syndrome that was not attributed to any dental or general condition. Interestingly, we observed that symptoms significantly improved or resolved with an adequate treatment and a long-term antireflux diet. A few studies investigated the involvement of reflux in dental lesions [17], or primary burning mouth syndrome [18,19,20], but authors reported conflicting results, which may be related to methodological discrepancies across studies [21]. Indeed, the majority of authors studied the association between burning mouth syndrome and reflux considering GERD and not LPR diagnostic criteria [18,19,20]. To date, it has been demonstrated that patients with LPR may not have GERD and vice versa [1]. The development of burning or pain mouth may be related to mucosal injury related to pepsin, which may be easily detected in saliva samples with peptest. Thus, the saliva pepsin detection could be useful to investigate the potential involvement of LPR in primary burning mouth syndrome, “idiopathic” aphthosis, or fissured tongue. 

Several studies demonstrated that reflux events may reach nasopharyngeal and nasal regions [22,23]. In this study, we identified patients who had nasal or otological findings associated with LPR, i.e., nasal obstruction, excessive nasopharyngeal mucus, or recurrent acute media otitis. The pharyngeal reflux events are known to be mainly gaseous, occurring upright and in daytime [24]. The occurrence of rhinopharyngeal reflux episodes may easily support the development of a reflux-related nasopharyngeal inflammation and the local production of sticky nasopharyngeal mucus, the obstruction of the Eustachian tube, and the development of otitis media disorders. Furthermore, pepsin has been identified in the secretion of otitis media in several studies [6,25,26]. According to nasal obstruction, two recent studies supported that LPR may lead to edema of the nasal mucosa, including the posterior part of the inferior turbinate, as observed in this study [27,28]. Interestingly, Magliulo et al. found pepsin in the tears [29], which may support the occurrence of a relationship between laryngopharyngeal reflux and tear disorders through the injury of the nasal mucosa of the inferior meatus.

The pepsin-related mucosal injury was initially studied in vocal fold tissues [3,30]. Pepsin may induce macroscopic and microscopic changes in the vocal fold mucosa, including epithelial cell dehiscence, microtraumas, inflammatory infiltrates, Reinke space dryness, mucosal drying, and epithelial thickening [31]. The development of severe dysplasia and its resolution after LPR treatment may probably support the potential impact of LPR in the development of some vocal fold morphological changes in nonsmoker patients. Clinically, LPR may have an impact on the clinical presentation and the therapeutic response of patients with asthma [8], which supports that the LPR-related inflammation may reach the bronchi. The observation of patients with LPR and chronic bronchitis that was not attributed to another disease supports the importance to keep in mind that LPR may be an irritative factor of the lower airway. In the same way, pepsin was found in the trachea and bronchi of patients with idiopathic stenosis [32].

In this case series, the association between LPR and atypical findings is possible but not proven. Despite the occurrence of an atypical and, therefore, different clinical presentation of LPR, the use of the term “LPR” has to be kept regarding the potential similar physiological mechanisms than classical LPR. According to our HEMII-pH analyses and previous studies [24], in both atypical and common clinical presentations of LPR, pharyngeal reflux events were gaseous and occurred in daytime and upright in the majority of patients. The detection of pepsin and other gastroduodenal enzymes in saliva, nasal, or bronchial secretions may form the basis for a future study and perhaps demonstrate the impact of LPR in the development of many unusual conditions. Gastroduodenal enzymes may irritate the upper aerodigestive tract mucosa but they may have an additional role on the local microbiota [33]. In the digestive area, much research has demonstrated the importance of gut bacteria in mucosa homeostasis, protection, recovery, or renewal [34,35]. Similarly, the critical role of microbiota was reported in respiratory tract diseases, such as tracheal stenosis or asthma [36,37]. Thus, it seems conceivable that LPR may impact the upper aerodigestive tract microbiota, leading to the development of some disorders.

The primary limitation of the present clinical study is the lack of tissue-related demonstrations of the involvement of reflux in the development of the atypical disorders. However, the occurrence of LPR at the HEMII-pH study and the complete resolution after treatment strongly support a clinical association. The retrospective design, the low number of included patients, and the short follow-up time of some patients are additional limitations of the study.

## 5. Conclusions

LPR may present with various clinical presentations, including mouth, eye, tracheobronchial, nasal, or laryngeal findings, which may all regress with an adequate treatment. Future studies are needed to better specify the relationship between LPR and these atypical findings through analyses identifying gastroduodenal enzymes in the enflamed tissue.

## Figures and Tables

**Figure 1 jcm-10-02439-f001:**
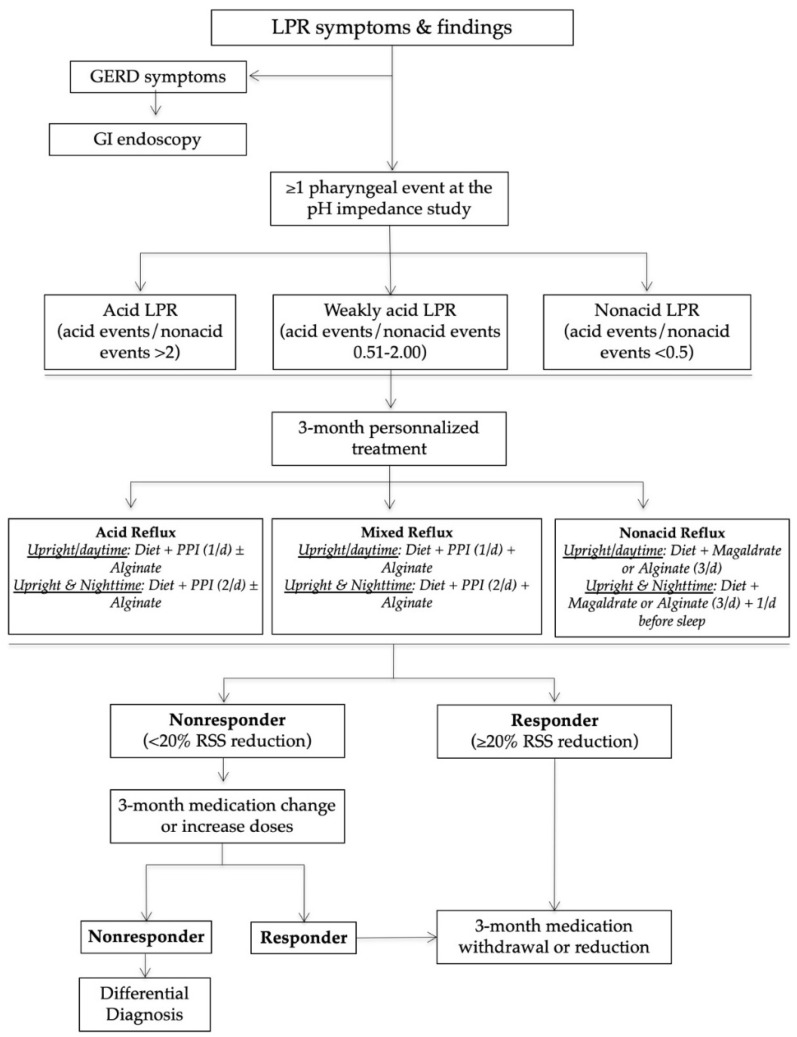
Management algorithm of LPR patients. Abbreviations: GERD = gastroesophageal reflux disease; GI = gastrointestinal; LPR = laryngopharyngeal reflux; PPI = proton pump inhibitor; RSS = reflux symptom score.

**Figure 2 jcm-10-02439-f002:**
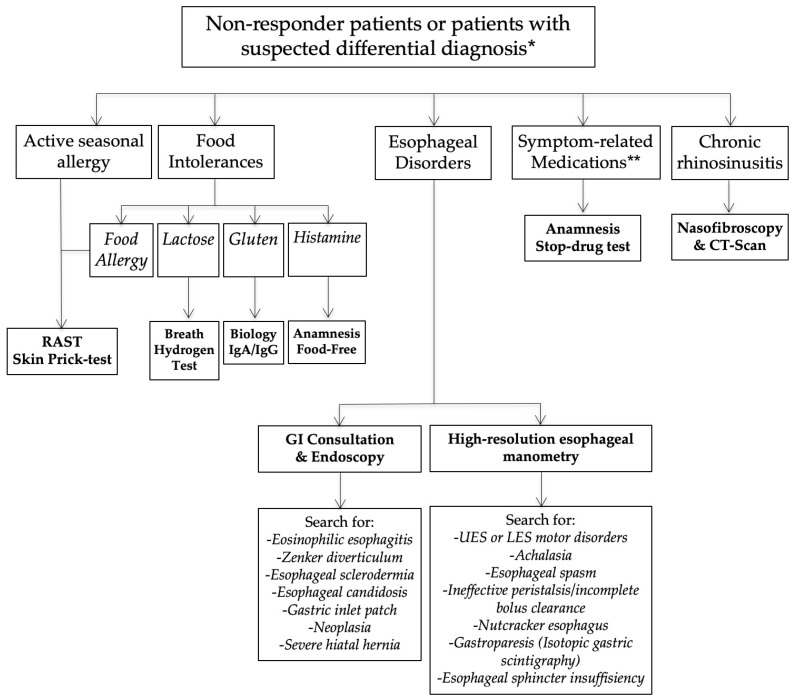
Nonresponder management. Abbreviations: GI = gastrointestinal; LES = lower esophageal sphincter; LPR = laryngopharyngeal reflux; UES = upper esophageal sphincter. * = differential diagnosis is a condition that may be associated with similar symptoms and findings. ** = Some symptoms may appear after the intake of medication (adverse events of drugs).

**Figure 3 jcm-10-02439-f003:**
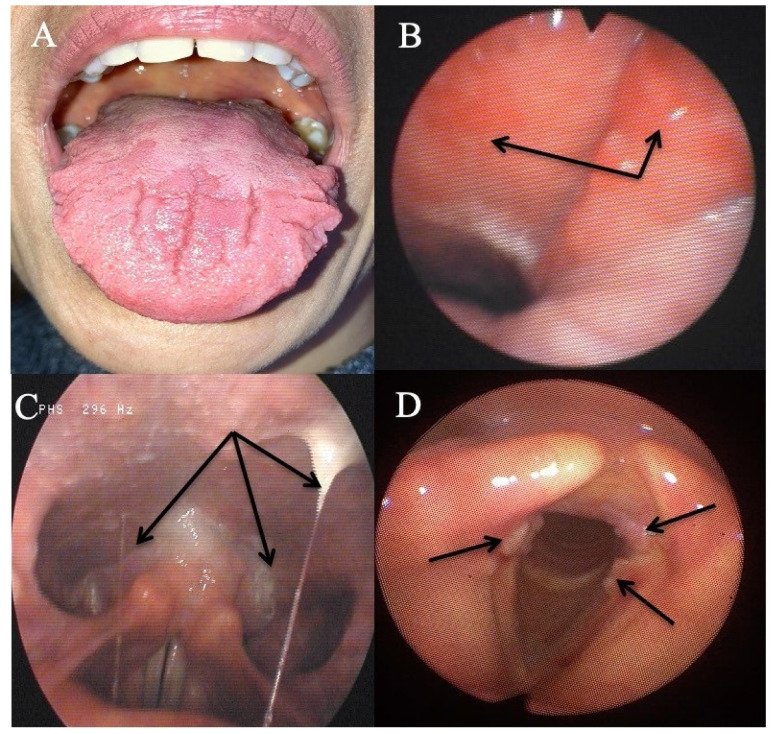
Some atypical findings associated with reflux. Fissured tongue (**A**), erythema of the nasopharynx and Eustachian meatus (**B**), sticky mucus from nasopharynx to oropharynx (**C**), and leukoplakia (**D**).

**Table 1 jcm-10-02439-t001:** Data of patients with oral manifestations.

PN	G	Age	Baseline Features		Post-Treatment Features			
			Atypical Presentation	HEMII-pH/RSS	Treatment	RSS/RSA	Presentation Evolution	Long-Term Follow-Up
1	M	36	Recurrent aphthosis and burning mouth	Upright acid reflux (58) *	Strict diet	RSS: 13	Resolution of aphthosis and	Long-term diet
			Dental check-up: normal	GI: normal	PPIs	RSA: 10	burning mouth	No recurrence (3-y)
				RSS: 48–RSA: 26	Magaldrate			
2	F	55	Recurrent aphthosis and burning mouth	Upright acid reflux (22) *	Strict diet	RSS: 5	Resolution of aphthosis and	Long-term diet
			Dental check-up: normal	GI: not performed	PPIs	RSA: NP	burning mouth	No recurrence (6-m)
				RSS: 50–RSA: 26	Alginate			
3	M	31	Mouth burning and severe anorexia	Upright nonacid reflux (11)	Histamine-free	RSS: 64	Resolution of pain and	Long-term histamine-free
			GI/dental check-up: normal	GI: normal	Diet	RSA: NP	weight gain	diet.
			Nutritionist: Histamine intolerance	RSS: 148–RSA: 14				No recurrence (1-y)
4	M	38	Tongue burning	Upright weakly acid reflux (15)	Strict diet	RSS: 16	Resolution of tongue	Long-term diet
			Dental check-up: normal	GI: normal	PPIs	RSA: 23	burning	No recurrence (1-y)
				RSS: 131–RSA: 37	Alginate			
5	M	55	Tongue burning	Upright acid reflux (27)	Strict diet	RSS: 48	Reduction of tongue	Long-term diet
			Dental check-up: normal	GI: GERD, hiatal hernia	PPIs	RSA: 21	burning	Long-term PPIs and
				RSS: 76–RSA: 24	Magaldrate			Magaldrate (1-y)
6	F	53	Tongue burning and fissured tongue	Upright acid reflux (19)	Strict diet	RSS: 135	Reduction of tongue	Long-term diet
			Dental check-up: normal	GI: GERD, esophagitis	PPIs	RSA: 22	Burning but no change	Long-term intermittent
				RSS: 247–RSA: 28	Magaldrate		in fissured tongue	Magaldrate (9-m)
7	F	54	Tongue and mouth burning	Upright acid reflux (38) *	Strict diet	RSS: 19	Resolution of tongue	Long-term diet
			Dental check-up: normal	GI: GERD	PPIs	RSA: 13	burning	No recurrence (3-y)
				RSS: 88–RSA: 26	Magaldrate			
8	F	62	Tongue and mouth burning	Upright acid reflux	Strict diet	RSS: 19	Resolution of tongue	Long-term diet
			Dental check-up: normal	GI: GERD, esophagitis	PPIs	RSA: 32	burning	One recurrence controlled
				RSS: 203–RSA: 22	Alginate			with alginate (3-y)
9	F	64	Tongue and mouth burning	Upright acid reflux (7)	Strict alkaline	RSS: 12	Resolution of tongue	Long-term diet
			Dental check-up: normal	GI: normal	Diet	RSA: NP	burning	No recurrence (6-m)
				RSS: 124–RSA: 32				
10	F	31	Recurrent burps and abdominal pain	Upright acid reflux (18)	Gluten-free	RSS: 40	Resolution of burps and	Long-term gluten-free
			GI check-up: gluten intolerance	GI: bulbitis	Diet	RSA: 17	Abdominal pain	diet.
				RSS: 167–RSA: 20				No recurrence (4-y)
11	F	36	Recurrent burps and abdominal pain	Upright nonacid reflux (2)	Lactose-free	RSS: 16	Resolution of burps and	Long-term lactosis-free
			GI check-up: lactose intolerance	GI: normal	Diet	RSA: 21	Abdominal pain	diet.
				RSS: 111–RSA: 31				No recurrence (4-y)

The number of pharyngeal reflux events is reported in brackets (column HEMII-pH/RSS), while the presence of GERD that was defined as a DeMeester score >14.72 or a length of time >4.0% of the 24 h recording spent below pH 4.0, is marked *. Abbreviations: G = gender; GERD = gastroesophageal reflux disease; GI = gastrointestinal; HEMII-pH = hypopharyngeal-esophageal multichannel intraluminal impedance pH-monitoring; m = month; NP = not provided; PN = patient number; PPI = proton pump inhibitor; RSA = reflux sign assessment; RSS = reflux symptom score; y = year.

**Table 2 jcm-10-02439-t002:** Data of patients with oral manifestations.

PN	G	Age	Baseline Features		Post-Treatment Features			
			Atypical Presentation	HEMII-pH/RSS	Treatment	RSS/RSA	Presentation Evolution	Long-Term Follow-Up
12	F	64	Resistant chronic nasal obstruction #	Upright weakly acid reflux (12)	Strict diet	RSS: 67	Resolution of nasal	Long-term diet
			Nasosinusal check-up: hypertrophy of	GI: normal	PPIs	RSA: NP	obstruction	No recurrence (6 months)
			the posterior part of the inferior turbine	RSS: 250–RSA: 29	Alginate			
13	F	50	Resistant chronic nasal obstruction #	Upright acid reflux (24)	Strict diet	RSS: 43	Resolution of nasal	Long-term diet
			Nasosinusal check-up: hypertrophy of	GI: not performed	PPIs	RSA: NP	obstruction	No recurrence (6 months)
			the posterior part of the inferior turbine	RSS: 58–RSA: 12	Alginate		Septoplasty not required	
14	F	66	Resistant chronic nasal obstruction	Upright weakly acid reflux (9) *	Strict alkaline	RSS: 184	Resolution of nasal	Long-term diet and alginate
			and tearing	GI: not performed	Diet	RSA: NP	obstruction and tearing	No recurrence (6-m) of
			Nasosinusal check-up: hypertrophy of	RSS: 210–RSA: 32			Resolution of edema of	tear and nasal symptoms
			the posterior part of the inferior turbine				inferior and middle meatus	Chronic throat symptoms
15	F	35	Recurrent suppurative media otitis and	Upright acid reflux (4) *	Strict alkaline	RSS: 2	Resolution of media	Long-term diet
			Ear pressure and pain	GI: not performed	Diet	RSA: 7	otitis	No recurrence (1-y) of
			Nasosinusal check-up: normal	RSS: 11–RSA: 20				symptoms or tympanic
			Otological check-up: retraction pocket					membrane findings.
16	M	37	Chronic media otitis	Upright acid reflux (3)	Strict alkaline	RSS: 48	Improvement of nasal	Long-term diet and short
			Nasosinusal check-up: obstruction and	GI: not performed	Diet	RSA: NP	obstruction	period of alginate
			erythema of the Eustachian tube. **	RSS: 73–RSA: 36				No recurrence (1-y)
17	M	36	Recurrent rhinopharyngitis/otitis	Upright weakly acid reflux (11)	Strict diet	RSS: 52	Resolution of rhino-	Long-term diet
			Otological check-up: normal	GI: not performed	PPIs	RSA: 20	pharyngitis posttreatment	No recurrence (6 months)
			Nasal check-up: controlled dust allergy	RSS: 107–RSA: 29	Alginate			

The number of pharyngeal reflux events is reported in brackets (column HEMII-pH/RSS), while the presence of GERD that was defined as a DeMeester score >14.72 or a length of time >4.0% of the 24 h recording spent below pH 4.0, is marked * in the same column. # Resistant chronic nasal obstruction = resistant to two types of nasal sprays (mometasone furoate and budesonide). ** Eustachian tube disorder highlighted with abnormal tympanometry and audiometry reporting retrotympanic membrane liquid. Abbreviations: F/M = female/male; G = gender; GERD = gastroesophageal reflux disease; GI = gastrointestinal; HEMII-pH = hypopharyngeal-esophageal multichannel intraluminal impedance pH-monitoring; m = month; NP = not provided; PN = patient number; PPI = proton pump inhibitor; RSA = reflux sign assessment; RSS = reflux symptom score; y = year. Broncho-laryngeal atypical manifestations.

**Table 3 jcm-10-02439-t003:** Broncho-laryngeal manifestations of reflux.

PN	G	Age	Baseline Features		Post-Treatment Features			
			Atypical Presentation	HEMII-pH/RSS	Treatment	RSS/RSA	Presentation Evolution	Long-Term Follow-Up
18	F	34	Severe idiopathic vocal fold dysplasia	Upright acid reflux (11)	Strict alkaline	RSS: 9	Resolution of dysplasia	Long-term diet
			Laryngeal check-up: normal	GI: not performed	Alginate	RSA: 20	within 6 months	No recurrence (6 months)
			No tobacco/toxic exposition history	RSS: 34–RSA: 39	Diet			
19	M	45	Severe idiopathic vocal fold dysplasia	Upright acid reflux (22)	Strict alkaline	RSS: 16	Resolution of dysplasia	Long-term diet
			Laryngeal check-up: normal	GI: not performed	Diet	RSA: 18	within 6 months	No recurrence (9 months)
			No tobacco/toxic exposition history	RSS: 73–RSA: 23				
20	M	38	Daily aspirations and pneumonia	Upright acid reflux (39) *	Strict alkaline	RSS: 81	Resolution of dysplasia	Long-term diet
			Lung/swallowing check-up: normal	GI: esophagitis	Diet	RSA: 18	within 6 months	No recurrence (6 months)
				RSS: 156–RSA: 27	Alginate			
21	F	65	Recurrent tracheobronchitis	Upright weakly acid reflux (34) *	Strict diet	RSS: 110	Resolution of tracheo-	Long-term diet
			Lung check-up: normal	GI: LES insufficiency	Magaldrate	RSA: 19	bronchitis within 6 months	Magaldrate (sometimes)
			No tobacco/asthma history	RSS: 415–RSA: 24				No recurrence (6 months)

The number of pharyngeal reflux events is reported in brackets (column HEMII-pH/RSS), while the presence of GERD that was defined as a DeMeester score >14.72 or a length of time >4.0% of the 24 h recording spent below pH 4.0, is marked *. Abbreviations: F/M = female/male; G = gender; GERD = gastroesophageal reflux disease; GI = gastrointestinal; HEMII-pH = hypopharyngeal-esophageal multichannel intraluminal impedance pH-monitoring; LES = lower esophageal sphincter; m = month; NP = not provided; PN = patient number; RSA = reflux sign assessment; RSS = reflux symptom score; y = year.

## Data Availability

Data may be available on request.

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
