# Peer review of "Atypical Clinical Presentation of Laryngopharyngeal Reflux: A 5-Year Case Series"

_jcm, 2021, doi:10.3390/jcm10112439_

Round 1
Reviewer 1 Report
Various cases are well documented as a study of cases with non-specific symptoms that may occur in laryngopharyngeal reflux disease. However, except the typical symptoms seen in LPRD, there have been many reports that various diseases with unusual symptoms occur in most areas connected to the pharynx, such as the oral cavity, nasal cavity, and middle ear that was mentioned in this paper.
Therefore, it is questionable whether defining atypical LPRD by the presence of atypical symptoms is meaningful and provide academic information.
Author Response
We agree. We proposed in the method section to define atypical symptoms and findings as symptoms or findings that are not reported in full version of Reflux Symptom Score and Reflux Sign Assessment. These two clinical instruments included the most prevalent symptoms and findings of LPR and their development were based on a systematic review of symptoms and findings associated with LPR (see doi: 10.1002/lary.27591.).
Methods: second paragraph: “The atypical LPR was defined as a clinical presentation with symptoms or findings that are not reported in reflux symptom score (RSS) [11] or reflux sign assessment (RSA) [12]. RSS is 22-item patient-reported outcome questionnaire that report the most prevalent otolaryngological, digestive and respiratory symptoms associated with LPR. RSA is a finding instrument including the most prevalent signs associated with LPR. Thus, the development of RSA was based on an initial observational study analyzing the prevalence of oral, laryngeal and pharyngeal findings associated with LPR in patients with a confirmed diagnosis (HEMII-pH). Both scores were developed after a systematic review of the most prevalent LPR symptoms and signs reported in the literature [13] and may be considered as a complete and reliable patient-reported outcome questionnaire or finding instrument [11].”

Reviewer 2 Report
The authors should be commended for their work on this manuscript.
Reference is made to Figures 1 and 2, but they are not included.
The numbering within Figure 3 should be changed to a-d instead of 1-4 - it is less confusing. Also, reference is made to Figure 3(2) before Figure3(1) in the manuscript - typically multiple references within one figure are presented in order in the manuscript.
Page 10, line 34 - reference needs to be added (currently says "XX")
Author Response
The authors should be commended for their work on this manuscript.
Reference is made to Figures 1 and 2, but they are not included.
We checked that the fig. 1 and 2 were included in the revised manuscript. Sorry for the lack of figures in the submitted version.
The numbering within Figure 3 should be changed to a-d instead of 1-4 - it is less confusing. Also, reference is made to Figure 3(2) before Figure3(1) in the manuscript - typically multiple references within one figure are presented in order in the manuscript.
We modified the figure 3 as requested by the reviewer. Moreover, we add two photos to the figure 3 to improve the iconography of the paper: edema of the back of the turbinate and edema of the inferior meatus. Please, check the new figure 3.
Page 10, line 34 - reference needs to be added (currently says "XX")
It’s done: “Interestingly, Magliulo et al. found pepsin in the tears [30], which may support the occurrence of a relationship between laryngopharyngeal reflux and tear disorders through the injury of the nasal mucosa of the inferior meatus. »

Reviewer 3 Report
This paper is a retrospective review of 351 patients with laryngopharyngeal reflux (LPR) over 4 years to identify the subgroup of 24 subjects with atypical symptoms. It appears the larger group was defined by the reflux symptom score (RSS) or reflux sign assessment (RSA). Atypical symptoms such as mouth burning or sinonasal complaints were confirmed to be associated with LPR if it was was confirmed with pH/impedance probe, if the finding resolved posttreatment and if the atypical finding of patient was not explained by another condition.
The paper expands definitions of LPR symptoms, takes a systematic approach to the hypothesis-driven study, and is well written (albeit requiring minor editing for English readability). The bigger issues are
1) LPR is already considered to be an "atypical" presentation of GERD, noting the same physiologic process of regurgitation but with different symptoms. Since this study looks at atypical LPR, then it's really just a more refined version of the same process. The danger would then be to call this another name which further claims a separate disease process when in fact it is all supraesophageal consequences of gastric regurgitation. This authors should address this comment in the introduction and discussion.
2) Diagnosing LPR solely by the European reflux symptom score (RSS) or reflux sign assessment (RSA) is limited by sensitivity and specificity in the absence of true heartburn or regurgitation. This is the same limitation as relying on a firm diagnosis using the Reflux Symptom Index and Reflux Findings Score published in the USA 20 years earlier. Ultimately the authors do require pH/impedance probes to confirm a diagnosis of reflux. As such, Table 1 should include the quantified number of episodes or %time in reflux documented on the HEMI-pH column.
There should be a title of the Discussion beginning on page 10. While there are multiple minor nuanced controversial comments, I would defer to the author's discretion. Please correct spelling in Table 3: "alkaline."
Author Response
- LPR is already considered to be an "atypical" presentation of GERD, noting the same physiologic process of regurgitation but with different symptoms. Since this study looks at atypical LPR, then it's really just a more refined version of the same process. The danger would then be to call this another name which further claims a separate disease process when in fact it is all supraesophageal consequences of gastric regurgitation. This authors should address this comment in the introduction and discussion.
GERD and LPR differed from some pathophysiological mechanisms, such as the nature and the time of the refluxate (gaseous & upright for LPR; often liquid and daytime and nighttime (supine) for GERD). According to DeMeester score (and definition of GERD), many patients with LPR have no GERD and vice versa. (ref. doi: 10.1002/lary.28736)
We specified these point in the introduction and we stated in the discussion that atypical LPR is still LPR and is not a new clinical disease.
Introduction: we added: “Gastroesophageal reflux disease (GERD) and LPR share some common pathophysiological mechanisms but may differ regarding the nature and time of occurrence of reflux events [1].”
Discussion: “Although the occurrence of atypical and, therefore, different clinical presentation of LPR, the use of the term “LPR” has to be kept regarding the potential similar physiological mechanisms than classical LPR. According to our HEMII-pH analyses and previous studies [25], in both atypical and common clinical presentations of LPR, pharyngeal reflux events were gaseous and occurred daytime and upright in the majority of patients.”
- Diagnosing LPR solely by the European reflux symptom score (RSS) or reflux sign assessment (RSA) is limited by sensitivity and specificity in the absence of true heartburn or regurgitation. This is the same limitation as relying on a firm diagnosis using the Reflux Symptom Index and Reflux Findings Score published in the USA 20 years earlier. Ultimately the authors do require pH/impedance probes to confirm a diagnosis of reflux. As such, Table 1 should include the quantified number of episodes or %time in reflux documented on the HEMI-pH column.
We added the number of pharyngeal reflux events.
There should be a title of the Discussion beginning on page 10. While there are multiple minor nuanced controversial comments, I would defer to the author's discretion. Please correct spelling in Table 3: "alkaline."
We corrected in all tables.

Round 2
Reviewer 2 Report
The authors made the suggested changes. However, I think the addition of the 2 pictures in Figure 3 (turbinate and meatus) don't really add to the manuscript and are difficult to interpret (i.e. they don't look that abnormal). I would remove these 2 pictures.
Author Response
Thank you for your comment.
We removed the 2 part of the figure as requested.
